# Inflammatory markers in world trade center workers with asthma: Associations with post traumatic stress disorder

**Juan P. Wisnivesky**[1,2], **Nikita Agrawal**[2]*, **Jyoti Ankam**[1], **Adam Gonzalez**[3], **Alex Federman**[1], **Steven B. Markowitz**[4], **Janette M. Birmingham**[5], **Paula J. Busse**[5]

**1** Division of General Internal Medicine, Icahn School of Medicine at Mount Sinai, NY, NY, United States of America, **2** Division of Pulmonary and Critical Care Medicine, Icahn School of Medicine at Mount Sinai, NY, NY, United States of America, **3** Barry Commoners Center for Health and Environment, Queens College, City University of NY, NY, NY, United States of America, **4** Department of Psychiatry and Behavioral Health, Stony Brook University, Stony Brook, NY, United States of America, **5** Division of Allergy and Immunology, Icahn School of Medicine at Mount Sinai, NY, NY, United States of America

* nikita.agrawal@mountsinai.org

**Data Availability Statement:** Data cannot be shared publicly because this project used data from the WTC Health Program General Responder

## Abstract

### Background

Post-traumatic stress disorders (PTSD) is associated with worse asthma outcomes in individuals exposed to the World Trade Center (WTC) site.

### Research question

Do WTC workers with coexisting PTSD and asthma have a specific inflammatory pattern that underlies the relationship with increased asthma morbidity?

### Study design and methods

We collected data on a cohort of WTC workers with asthma recruited from the WTC Health Program. Diagnosis of PTSD was ascertained with a Structured Clinical Interview for DSM-5 (Diagnostic and Statistical Manuel of Mental Disorders) and the severity of PTSD symptoms was assessed with the PTSD Checklist 5. We obtained blood and sputum samples to measure cytokines levels in study participants.

### Results

Of the 232 WTC workers with diagnosis of asthma in the study, 75 (32%) had PTSD. PTSD was significantly associated with worse asthma control (p = 0.002) and increased resource utilization (p = 0.0002). There was no significant association (p>0.05) between most blood or sputum cytokines with PTSD diagnosis or PCL-5 scores both in unadjusted and adjusted analyses.

### Interpretation

Our results suggest that PTSD is not associated with blood and sputum inflammatory markers in WTC workers with asthma. These findings suggest that other mechanisms

Data Center at the Icahn School of Medicine at Mount Sinai. Because of regulations from the parent study we are not able to share data, even if de-identified. Rather, access to data would have to go through established World Trade Center Health Program General Responder Data Center mechanisms. The contact person for data requests is Andrew C. Todd, PhD at Mount Sinai Icahn School of Medicine (andrew.todd@mssm.edu).

**Funding:** JPW received funding by the Centers for Diseases and Control and Prevention/National Institute for Occupational Safety and Health (U01OH011312). The funders did not play any role in the study design, data collection, analysis, decision to publish or preparation of the manuscript.

**Competing interests:** JPW received honorarium from Banook, PPD, Atea Pharmaceutical, and Prospero and research grants from Regeneron, Sanofi, Axella, and Arnold Consulting. PB has received consulting fees from CSL Behring, Takeda, Kalvista, BioCryst, CVS Specialty, Regeneron and research funding from CSL Behring, Takeda, and Kalvista. Other authors declare no conflict of interest. All other authors have no competing interests to declare.

**Abbreviations:** PTSD, Post-traumatic Stress Disorder; WTC, World Trade Center; IL, Interleukin; IFN, Interferon; TNF, tumor-necrosis factor; WTCHP, World Trade Center Health Programs; ACQ, Asthma Control Questionnaire; AQLQ, Asthma Quality of Life Questionnaire; SCID-5, Structured Clinical Interview for Diagnostic and Statistical Manual for Mental Disorders Version Five; PCL-5, Post-traumatic Stress Disorder Checklist 5.

likely explain the association between PTSD and asthma control in WTC exposed individuals.

## Introduction

Asthma and post-traumatic stress disorder (PTSD) are the most common chronic conditions in World Trade Center (WTC) rescue and recovery workers, each affecting approximately 30% of individuals [1]. A large number of local residents and passersby also suffer from these conditions [2, 3]. Moreover, a great percentage of these individuals continue to report uncontrolled asthma (defined as increased daytime and nighttime respiratory symptoms and frequent rescue medication use) and have increased acute asthma-related healthcare resource utilization several years after exposure to the WTC pile [4, 5]. Thus, asthma remains a major source of morbidity and compromised quality of life among the WTC-exposed populations.

Several studies have shown a very strong relationship between PTSD and increased asthma morbidity in the WTC worker population, including worse disease control, functional impairment, increased urgent outpatient visits, emergency room (ER) visits and hospitalizations, and poorer quality of life [6, 7]. PTSD may contribute to worse asthma morbidity in WTC workers via systemic inflammation that can be seen in this condition. Specifically, patients with PTSD have increased levels of interleukin (IL)-1α, IL-2, IL-6, and tumor necrosis factor-alpha (TNF-α) [8, 9]. The enhanced inflammation modulated by various cytokines in PTSD thus may worsen asthma.

Major asthma endotypes, which describe asthma subtypes based on inflammatory mechanisms include eosinophilic and non-eosinophilic asthma [10–13]. Eosinophilic asthma is characterized by eosinophilia driven by $TH_2$ dominant inflammation. Non-eosinophilic asthma is characterized by neutrophilia driven by key cytokines, including IL-1a, IL-6 and IL-17, which are also elevated in PTSD [14, 15]. Therefore, systemic inflammatory changes associated with PTSD, may worsen asthma outcomes by driving a neutrophilic asthma phenotype.

However, there are limited data evaluating inflammatory markers in WTC workers with asthma and PTSD and those without PTSD. This is an important knowledge gap preventing a potentially more personalized asthma treatment approach for these individuals. In this study, we used data from a cohort of WTC workers with asthma to assess the potential differences in plasma and sputum inflammatory markers among individuals with vs. without comorbid PTSD. We hypothesized that in WTC workers with asthma, the presence of PTSD would be linked to increased expression of cytokines associated with non-eosinophilic asthma.

## Methods

### Study population

We enrolled a cohort of WTC workers selected from the Mount Sinai Hospital and Northwell Health System locations of the WTC Health Programs (WTCHP) Clinical Centers of Excellence. To be eligible to participate in the WTCHP, individuals must have worked or volunteered at the WTC site for ≥4 hours from September 11 to 14, 2001, ≥24 hours during September 2001, or ≥80 hours from September to December 2001 [16]. The health program also provides care for workers from the Office of the Chief Medical Examiner and Port Authority Trans Hudson Corporation who participated in recovery activities. Study participants were ≥18 years of age, had a physician diagnosis of asthma, and spoke English or Spanish. Exclusion criteria included a prior diagnosis of chronic obstructive lung disease (COPD)

or other chronic lung disease, or history of ≥15 pack-years of tobacco smoking, to avoid including WTC workers with undiagnosed COPD.

We sent opt in recruitment letters to potential participants from February 2017 to January 2020 and then called them by phone to offer enrollment and to conduct a brief eligibility assessment. WTC workers who agreed to participate in the study were invited to an in-person interview conducted by trained research coordinators in English or Spanish based on participant's preferences. All participants signed a written informed consent, and the study was approved by Institutional Review Board of the Icahn School of Medicine at Mount Sinai (IRB # 16–01055) and Queens College, City University of New York (IRB# DOE000652). Identifiable patient information was de-identified into study numbers and identifiable patient information was limited to JA and JB for this study.

## Study variables

All study variables were collected during the recruitment period of February 2017 to January 2020. The survey instrument included standardized questions adapted from the National Health Interview Survey assessing participant's sociodemographic characteristics (e.g., age, sex, race, ethnicity, education, and income) [17]. We collected data on age of asthma onset, diagnosis in relation to WTC exposure (pre- vs. post- 9/11/2001), history of allergies, and other comorbidities (e.g., chronic sinusitis, allergic rhinitis, gastroesophageal reflux) that may impact asthma control, asthma medications, and history of acute resource utilization (including ER visits and hospitalizations) in the past 12 months. In addition to health care resource utilization, measures of asthma control included responses to the Asthma Control Questionnaire (ACQ) and the Mini-Asthma Quality of Life Questionnaire (AQLQ). The ACQ is a 6-item validated questionnaire with individual items ranging from 0 (totally controlled) to 6 (severely uncontrolled), with a mean score of ≥1.5 indicating poorly controlled asthma [18, 19]. The AQLQ is a 15-item validated questionnaire to assess the impact of asthma on quality of life with higher scores indicating better quality of life [20]. A difference of 0.5 units is considered clinically meaningful for both scales [21].

We collected information on coexisting conditions including the history of physician diagnoses of hypertension, diabetes, and coronary artery disease. The extent of WTC exposure was categorized based on prior criteria developed using data on the total amount of time spent at the WTC site, the level of exposure to the WTC cloud, and history of working on the pile [1]. Using these data, we classified WTC workers as having experienced none, mild, moderate or severe exposure.

Presence or absence of PTSD was determined using the Structured Clinical Interview for Diagnostic and Statistical Manual for Mental Disorders Clinician Version Five (SCID-5) [22, 23]. We selected the SCID-5 to identify participants with PTSD as this tool is well-established as the gold standard for psychiatric diagnosis in clinical research. The SCID-5 follows diagnostic criteria based on the Diagnostic and Statistical Manuel of Mental Disorders 5th edition (DSM-5), is available in English and Spanish, and is well validated. An experienced mental health specialist trained the research staff (who had prior training in psychology) who conducted the SCID-5 interviews according to established procedures. Additionally, we used the PTSD Checklist 5 (PCL-5), a reliable self-report tool also validated in English and Spanish, to assess for and quantify the severity of PTSD symptoms [24, 25].

Peripheral blood was collected in EDTA tubes to measure plasma cytokine levels. After collection, the plasma was separated via centrifugation (1400G) for 10 min at 4˚C. The plasma supernatant was then collected and stored at -80˚C until further processing for cytokine protein level. Sputum was induced adopting methodology used by the National Heart, Lung and

Blood Institute-sponsored asthma networks [26]. Briefly, participants performed baseline spirometry and then received 360 micrograms of short-acting bronchodilator prior to induction. Subjects received nebulized hypertonic saline (3%) for 12 minutes, over three 4-minute intervals to induce sputum. Sputum was processed by the whole sputum method [27]. Cytospins were prepared and stained with Diff-Quick (Dade Diagnostics of PR, Aguada, PR) for differential cell counts. The sputum cell differential was determined by counting at least 500 white blood cells on the cytospin slides, excluding samples with a cell viability <50% or >20% squamous cells. All sputum samples were read by the same blinded reader (J.B.) to limit potential variability in slide interpretation. Sputum supernatant and plasma were assayed for a panel of cytokines using a multiplex assay (Milliplex Human Th17 Magnetic Bead, Eotaxin-1, TGF-α and IL-8, Billerica, MA) according to manufacturer's instructions (which recommended using non-diluted samples). Briefly, samples, standards, and controls were added to the appropriate wells. The premixed magnetic beads were added to each well and incubated on a plate shaker for 16 to 18 hours at 4°C. After washing, detection antibody was added and incubated on a plate shaker for 1 hour at room temperature, followed by Streptavidin-Phycoerythrin for 30 minutes. The plate was run on the Luminex 200 system (Luminex Corp., Austin, TX) and data analyzed using the MILLIPLEX Analyst Software (EMD Millipore Corp., Billerica, MA). In samples where the cytokine concentration was below the lower limit of detection, it was not possible to determine whether the true cytokine was zero, so the concentration was assigned to be half the lower limit of detection [28]. None of the cytokine samples were above the upper limit of detection, and therefore were included in the analyses.

For this study, we examined both blood and sputum samples as prior research has demonstrated both samples provide information on systemic inflammation in asthma [10]. While evaluation of sputum markers are considered an optimal measure of local inflammation, it is often more difficulty to obtain appropriate samples, particularly in the context of the COVID-19 pandemic. Thus, serum samples were additionally obtained.

## Statistical analysis

We compared the baseline characteristics of WTC workers with and without PTSD (based on SCID-5) using a t-test, Wilcoxon test, or chi-square test, as appropriate. The unadjusted association of PTSD with blood and sputum cytokines levels and sputum cell counts was evaluated using a Wilcoxon test. The unadjusted relationship between PCL-5 scores and cytokine levels were calculated using the Spearman correlation coefficient. We used linear regression models to assess the adjusted association of PTSD with blood and sputum cytokines or sputum cell counts after controlling for age, sex, race/ethnicity, income, WTC exposure, smoking history, asthma history and regimen, and depression. Similar analyses were used to evaluate the adjusted association of PCL-5 scores with blood and sputum inflammatory markers or sputum cell counts.

We conducted sample size calculations that showed that with 230 subjects, the study had >80% power to determine if PTSD was associated with an effect size of 0.4 in cytokine levels assuming a prevalence of PTSD of approximately 30% in the study cohort. Analyses were performed with SAS software version 9.4 (SAS, Cary, NC) using two-sided p-values at a 0.05 significant level.

## Results

The study was conducted between February 2017 and January 2020. Workers enrolled in the Mount Sinai Hospital and Northwell System sites of the WTCHP program who had agreed to participate in new research studies were contacted. Of these, 177 WTC workers were found to

be ineligible during screening (41 [23%] had no history of asthma, 40 [23%] had comorbid COPD, 25 [14%] were not English or Spanish speakers, and 71 [40%] due to other reasons). Of the 360 WTC workers consented into the study, 3 (1%) participants were found to be ineligible after enrollment, 13 (4%) withdrew after recruitment, 105 (29%) were excluded due to lack of blood cytokine measures and 7 (2%) were excluded because they did not undergo the SCID-5 assessment to determine PTSD status. Thus, our final study cohort consisted of 232 WTC workers with asthma. Of these, 76 completed the sputum induction visit.

Based on SCID-5 results, 75 (32%) participants had current PTSD. The baseline characteristics of WTC workers with and without PTSD are shown in Table 1. WTC workers with PTSD were more likely to have an income ≤$3,000 per month (p<0.0001); other baseline sociodemographic characteristics, asthma history, and comorbidities were not significantly different across the two study groups (p>0.05 for all comparisons). PTSD was significantly associated with worse asthma control (p = 0.002), higher rates of ER visits (p = 0.0002) and hospitalizations (p = 0.03) and poorer asthma-related quality of life (p<0.0001).

Table 2 shows the unadjusted association of PTSD diagnosis with blood and sputum cytokines. Overall, there were no significant associations between PTSD diagnosis and any of the blood cytokines assessed in the study (p>0.05 for all comparisons). For sputum cytokines, IL-5 was lower (1.7±1.5 vs. 1.0±1.0, p = 0.03) in WTC workers with PTSD vs. those without PTSD, respectively. Additionally, neutrophils were significantly higher in the sputum cell differential (19.1% vs. 8.8%, p = 0.047) in WTC workers with PTSD; there were no significant differences in the distribution of other cell types (p>0.05 for all comparisons). In secondary analyses, we found that the severity of PTSD symptoms (PCL-5 scores) was not significantly correlated with blood (p<0.05 for all comparisons) or sputum (p<0.05 for all comparisons) cytokine levels. Similarly, there was no significant correlation between PCL-5 scores and sputum cell differential counts (p>0.05 for all comparisons; S1 Table in S1 Appendix).

Adjusted analyses showed no significant association between PTSD diagnosis and blood cytokine levels or sputum cytokine levels (Table 3). There was no significant difference in the adjusted cell count differential of WTC workers with vs. without PTSD. However, the severity of PTSD symptoms was significantly associated with differences in blood levels of IL-1β (β coefficient: 0.03 units per 1 unit increase in PCL-5 score, 95% CI: 0.01–0.06), IL-2 (β coefficient: 0.03 units per 1 unit increase in PCL-5 scores, 95% CI: 0.01–0.06), IL-7 (β coefficient: 0.01 units per 1 unit increase in PCL-5 score, 95% CI: 0.02–0.01), IFN-gamma (β coefficient 0.3 units per 1 increase in PCL-5 score, 95% CI: 0.05–0.6), TNF-alpha (β coefficient 0.1 units per 1 increase in PCL-5 score, 95% CI: 0.02–0.2) and VEGF (β coefficient: 1.4 units per 1 unit increase in PCL-5 score, 95% CI: 0.6–2.3; S2 Table in S1 Appendix). There were no significant associations of PTSD symptoms with sputum cytokine levels or sputum cell count differential in adjusted analyses.

## Discussion

Asthma and PTSD are among the most prevalent physical and mental comorbidities among WTC workers [1]. While PTSD is one of the strongest risk factors for increased asthma morbidity and poor quality of life in individuals exposed to the dust at the WTC site [29, 30], the underlying mechanism for this association is not well understood. It has been hypothesized that the airway inflammation seen in asthma may be modulated by the systemic inflammation observed in PTSD [31]. Furthermore, given that non-eosinophilic asthma, which is characterized by neutrophilia, is driven by cytokines seen in PTSD including IL-1α, IL-6 and IL-17, we hypothesized that individuals with asthma and PTSD would have distinct inflammatory markers compared to those without PTSD. Understanding inflammatory phenotypes in asthma is

**Table 1. Baseline characteristics of world trade center workers with asthma according to post traumatic stress disorder diagnosis.**

| Characteristic | No PTSD N = 157 | PTSD N = 75 | P-value |
|---|---|---|---|
| **Demographic and Exposure Factors** | | | |
| Age (Years), Mean (SD) | 56.1 (7.6) | 54.8 (9.5) | 0.25 |
| Female, No. (%) | 34 (22) | 21 (28) | 0.29 |
| Race/Ethnicity, No. (%) | | | |
| White | 58 (37) | 20 (27) | 0.11 |
| Black | 34 (22) | 11 (15) | |
| Hispanic | 46 (29) | 31 (41) | |
| Other | 19 (12) | 13 (17) | |
| Married, No. (%) | 100 (66) | 37 (54) | 0.08 |
| Education, No. (%) | | | |
| High school of less | 40 (26) | 19 (28) | 0.85 |
| Some college or college graduate | 112 (74) | 50 (72) | |
| Income, No. (%) | | | |
| ≤$3,000 per month | 32 (22) | 32 (52) | <0.0001 |
| >$3,000 per month | 114 (78) | 29 (48) | |
| WTC Exposure, No. (%) | | | |
| Low | 11 (11) | 4 (8) | 0.41 |
| Intermediate | 69 (67) | 28 (58) | |
| High | 10 (10) | 9 (19) | |
| Very high | 12 (12) | 7 (15) | |
| Smoking History, No. (%) | | | |
| Never | 111 (77) | 54 (77) | 0.94 |
| Former | 31 (22) | 15 (22) | |
| Current | 1 (1) | 1 (1) | |
| **Asthma-related Factors** | | | |
| Post 9/11 Asthma, No. (%) | 120 (91) | 59 (86) | 0.24 |
| Sensitized to Aeroallergens (at least one), No. (%) | 88 (83) | 42 (88) | 0.48 |
| History of Intubation, No. (%) | 2 (1) | 3 (4) | 0.18 |
| On Asthma Controller Medication, No. (%) | 136 (87) | 62 (83) | 0.42 |
| Asthma Control, No. (%) | | | |
| Well controlled | 60 (39) | 15 (21) | 0.002 |
| Uncontrolled | 31 (20) | 9 (13) | |
| Very poorly controlled | 62 (41) | 46 (66) | |
| Poor Asthma-Related Quality of Life, No. (%) | 41 (27) | 39 (56) | <0.0001 |
| Hospitalized for Asthma in the Past Year, No. (%) | 3 (2) | 6 (8) | 0.03 |
| Emergency Room Visit for Asthma in the Past Year, No. (%) | 14 (9) | 20 (29) | 0.0002 |
| Gastric Esophageal Reflux Disorder | 108 (69) | 50 (67) | 0.74 |
| Chronic Sinusitis | 90 (57) | 41 (55) | 0.70 |
| Diabetes | 27 (17) | 13 (17) | 0.98 |
| Hypertension | 67 (43) | 30 (40) | 0.70 |

PTSD: Post Traumatic Stress Disorder, WTC: World Trade Center

key as it influences management (i.e., eosinophilic asthma is more corticosteroid responsive than non-eosinophilic asthma). However, we did not find significant differences in the levels of blood or sputum cytokines in WTC workers with asthma with vs. without PTSD in adjusted analyses. These findings suggest that other mechanisms, like symptom perception, mental

**Table 2. Unadjusted associations between post-traumatic stress disorder with serum and sputum cytokine levels in world trade center workers with asthma.**

| Cytokine | Blood | | | Sputum | | |
|---|---|---|---|---|---|---|
| | No PTSD Mean (SD) | PTSD Mean (SD) | P-value | No PTSD Mean (SD) | PTSD Mean (SD) | P-value |
| IL-1α | 107.6 (340.8) | 97.5 (306.2) | 0.63 | 102.2 (138.2) | 115.2 (127.8) | 0.66 |
| IL-1β | 0.6 (1.1) | 1.1 (2.6) | 0.35 | 15.6 (23.4) | 13.1 (11.5) | 0.63 |
| IL-2 | 0.5 (0.9) | 1 (2.3) | 0.41 | 0.2 (0.6) | 0.1 (0.3) | 0.88 |
| IL-3 | 0.6 (0.6) | 0.7 (0.6) | 0.25 | 0.07 (0.2) | 0.1 (0.3) | 0.84 |
| IL-4 | 141.6 (584.0) | 165.9 (529.8) | 0.35 | 12.5 (14.6) | 12.2 (18.9) | 0.55 |
| IL-5 | 3 (8.7) | 3.7 (7.9) | 0.54 | 1.7 (1.5) | 1.0 (1.0) | 0.03 |
| IL-6 | 14.9 (52.1) | 14.2 (39.1) | 0.72 | 14.3 (20.3) | 17.4 (13.3) | 0.05 |
| IL-7 | 1 (2.8) | 2.4 (6.5) | 0.42 | 13.8 (10.6) | 13.5 (9.2) | 0.97 |
| IL-8 | 8.4 (21.7) | 9.3 (19.3) | 0.84 | 1204.5 (1265.2) | 992.0 (1306.7) | 0.43 |
| IL-10 | 6.6 (17.4) | 5.7 (12.7) | 0.57 | 3.2 (2.8) | 4.2 (4.5) | 0.59 |
| IL-12 | 11.3 (19.4) | 12.3 (27.4) | 0.81 | 1.8 (5.9) | 1.2 (2.1) | 0.74 |
| IL12p70 | 5.2 (28.7) | 3.1 (6.4) | 0.34 | 3.5 (7.5) | 2.6 (1.9) | 0.93 |
| IL-13 | 30.7 (118.2) | 30.0 (93.5) | 0.70 | 4.7 (3.4) | 3.7 (2.4) | 0.51 |
| IL-15 | 2.4 (4.2) | 2.1 (3.5) | 0.73 | 3.5 (4.4) | 3.2 (2.4) | 0.60 |
| IL-17α | 3.6 (13.9) | 4.9 (18.6) | 0.46 | 1.2 (2.7) | 0.7 (1.1) | 0.42 |
| IL-1Rα | 19.5 (29.0) | 22.4 (31.9) | 0.55 | 4119.6 (2321.1) | 3864.8 (2616.7) | 0.52 |
| G-CSF | 22.1 (22.8) | 26.4 (31.7) | 0.92 | 236.1 (415.9) | 407.2 (518.9) | 0.20 |
| GM-CSF | 4.5 (7.0) | 5.3 (8.3) | 0.78 | 1.5 (1.2) | 1.2 (0.8) | 0.65 |
| IP10 | 468.2 (446.7) | 438.4 (293.6) | 0.94 | 3023.7 (3781.5) | 3843.1 (4045.1) | 0.18 |
| IFN-α2 | 17.7 (26.0) | 18 (30.5) | 0.67 | 11.6 (8.7) | 12.2 (11.3) | 0.95 |
| IFN−γ | 4.3 (8.3) | 8.9 (30.6) | 0.94 | 2.3 (2.1) | 2.5 (2.7) | 0.64 |
| Eotaxin | 118.5 (82.0) | 135.6 (90.7) | 0.24 | 19.2 (24.5) | 16.3 (15.5) | 0.68 |
| EGF | 25.1 (28.7) | 19.7 (19.6) | 0.26 | 546.6 (394.5) | 441.6 (173.6) | 0.65 |
| MCP1 | 374.8 (186.6) | 383.7 (176.7) | 0.58 | 971.7 (1010.5) | 626.0 (387.0) | 0.38 |
| MIP1-α | 4.9 (16.0) | 4.7 (12.2) | 0.53 | 27.0 (53.0) | 27.3 (29.0) | 0.80 |
| MIP1-β | 14.6 (15.1) | 13.6 (16.3) | 0.50 | 42.3 (104.7) | 34.2 (48.4) | 0.57 |
| RANTES | 3066.1 (2254.7) | 2688.8 (1654.1) | 0.62 | 7.3 (5.5) | 8.1 (7.4) | 0.85 |
| TNF-α | 9.1 (5.7) | 10.8 (7.8) | 0.21 | 5.8 (10.1) | 5.4 (4.9) | 0.64 |
| TNF-β | 59.6 (221.7) | 51.2 (153.5) | 0.97 | 0.5 (2.1) | 0.1 (0.4) | 0.37 |
| VEGF | 23.6 (34.9) | 39.3 (85.5) | 0.66 | 665.6 (926.1) | 522.0 (377.8) | 0.51 |
| **Cell Count Differential** | | | | Percentage (SD) | Percentage (SD) | P-value |
| Eosinophils (%) | - | - | - | 3.7 (4.0) | 6.8 (8.9) | 0.17 |
| Neutrophils (%) | - | - | - | 8.8 (7.2) | 19.1 (17.5) | 0.047 |
| Macrophages (%) | - | - | - | 86.6 (9.0) | 73.3 (23.9) | 0.14 |
| Lymphocytes (%) | - | - | - | 0.9 (0.7) | 0.8 (0.5) | 0.64 |

The range of detection for the assay is 0–10,000 pg/ml for all cytokines

PTSD: Post Traumatic Stress Disorder, IL: interleukin, G-CSF: granulocyte colony stimulating factor, GM-CSF: granulocyte-macrophage colony-stimulating factor, IP: interferon gamma inducible protein, IFN: interferon, EGF: epidermal growth factor, MIP: macrophage inflammatory protein, TNF: tumor necrosis factor, VEGF: vascular endothelial growth factor

health comorbidities, or social and behavioral factors, may explain the relation between PTSD and worse asthma control in WTC workers [32, 33].

Prior studies have examined inflammatory pathways in PTSD to identify markers that are overexpressed in patients with asthma [31]. One study examining combat veterans with PTSD demonstrated an increase in peripheral blood mononuclear cells with an increase in absolute number of lymphocyte subsets compared to control individuals. In this study, among

**Table 3. Adjusted mean differences in blood and sputum cytokine level in world trade center workers with vs. without post-traumatic stress disorder.**

| Cytokine | Blood | | Sputum | |
|---|---|---|---|---|
| | Mean Difference | 95% Confidence Interval | Mean Difference | 95% Confidence Interval |
| IL-1α | -46.4 | -196.9 to 104 | -0.4 | -122.3 to 121.6 |
| IL-1β | 0.4 | -0.5 to 1.3 | 0.6 | -17.6 to 18.9 |
| IL-2 | 0.4 | -0.3 to 1.2 | -0.3 | -0.8 to 0.3 |
| IL-3 | 0.1 | -0.2 to 0.3 | -0.08 | -0.3 to 0.1 |
| IL-4 | 46.6 | -269.6 to 362.9 | -3.1 | -10.6 to 4.4 |
| IL-5 | -0.006 | -4.2 to 4.2 | -0.9 | -2.3 to 0.4 |
| IL-6 | -3.3 | -27.7 to 21.2 | 15.7 | -4.3 to 35.6 |
| IL-7 | 1.6 | -0.2 to 3.6 | -0.5 | -10.1 to 9.1 |
| IL-8 | -0.2 | -10.6 to 10 | 388.0 | -1004.7 to 1780.7 |
| IL-10 | -0.4 | -7.5 to 6.7 | 0.6 | -3.1 to 4.3 |
| IL-12 | -5.1 | -14.5 to 4.3 | -1.5 | -7.5 to 4.4 |
| IL12p70 | -1.2 | -14.8 to 12.3 | -1.9 | -8.2 to 4.4 |
| IL-13 | -11.4 | -63.8 to 41.1 | -2.2 | -5.6 to 1.2 |
| IL-15 | -0.5 | -2.6 to 1.6 | -0.6 | -4.8 to 3.6 |
| IL-17α | 1.5 | -6.8 to 9.8 | 0.03 | -2.7 to 2.8 |
| IL-1Rα | 4.4 | -9.9 to 18.8 | -530.7 | -2839.9 to 1778.5 |
| G-CSF | 6.9 | -4.7 to 18.5 | 313.5 | -135.3 to 762.4 |
| GM-CSF | 0.8 | -3.0 to 4.6 | -0.3 | -1.5 to 0.9 |
| IP10 | -114.4 | -310.1 to 81.2 | 1445 | -2186.2 to 5076.2 |
| IFN-α2 | -4.4 | -17.1 to 8.2 | -3.2 | -10.8 to 4.5 |
| IFN–γ | 4.0 | -5.5 to 31.5 | 0.1 | -2.4 to 2.7 |
| Eotaxin | -3.5 | -38.5 to 31.4 | -0.8 | -24.7 to 23 |
| EGF | -7.1 | -19.6 to 5.4 | -33.6 | -348.4 to 281 |
| MCP1 | -12.2 | -91.8 to 67.3 | -322.8 | -1223.1 to 577.5 |
| MIP1-α | -2.5 | -9.5 to 4.6 | 4.7 | -48.4 to 57.9 |
| MIP1-β | -1.3 | -7.6 to 4.9 | 2.0 | -102.1 to 106.1 |
| RANTES | -490.4 | -1430.7 to 449.5 | 2.2 | -3.6 to 8.1 |
| TNF-α | 2.0 | -0.8 to 4.8 | 2.5 | -7.1 to 12.2 |
| TNF-β | - 15.6 | -112.7 to 81.4 | -0.5 | -2.6 to 1.5 |
| VEGF | 14.6 | -15.7 to 44.9 | -172.9 | -964.2 to 618.3 |
| **Cell Count Differential** | | | Mean Difference | 95% Confidence Interval |
| Eosinophils (%) | - | - | 0.7 | -6.0 to 7.3 |
| Neutrophils (%) | - | - | 0.1 | -18.1 to 18.2 |
| Macrophages (%) | - | - | -0.8 | -21.8 to 20.1 |
| Lymphocytes (%) | - | - | 0.1 | -1.2 to 1.4 |

IL: interleukin, G-CSF: granulocyte colony stimulating factor, GM-CSF: granulocyte-macrophage colony-stimulating factor, IP: interferon gamma inducible protein, IFN: interferon, EGF: epidermal growth factor, MIP: macrophage inflammatory protein, TNF: tumor necrosis factor, VEGF: vascular endothelial growth factor

lymphocyte subclasses, while there was no significant increase in Th17 cells, there did appear to be a correlation with Th17 cells and PTSD severity. Additionally, Th2 cells remained unchanged with no correlation in PTSD scores; however, there was a statistically significant increase in Th1 cells that also positively correlated with increases in PTSD severity [34]. Expanding on these findings, clinical studies have found a lack of association of other Th2 driven conditions such as allergic disorders and hay fever in individuals with vs. without PTSD [35]. Beyond lymphocyte subclasses, multiple studies have demonstrated that

individuals with PTSD have altered levels of IL-1β, IL-6, TNF-alpha and IFN-gamma compared to those without PTSD [31].

In our cohort of WTC workers with asthma, adjusted analyses did not show differences in blood or sputum cytokine levels in individuals with vs. without PTSD. Moreover, our study did not find significant differences in sputum cell differential counts after adjusting for potential confounders. Our findings suggest that PTSD diagnosis is not linked with increased asthma morbidity through a pathway involving specific inflammatory markers. Moreover, in WTC workers with asthma, the presence of PTSD is not correlated with a single endotype (eosinophilic vs. non-eosinophilic) of asthma. Consistent with prior studies, our secondary analyses showed an association of the severity of PTSD symptoms with blood levels of IL-1B, IL-7, IFN gamma and TNF- α [8]. These cytokines are notable as they have also been implicated in non-eosinophilic asthma for which treatments such as inhaled corticosteroids and biologics targeting eosinophilic inflammation are classically less efficacious. However, these findings were limited to secondary analyses and were not replicated in comparisons of sputum cytokines. Thus, additional research is needed to determine if the severity of PTSD may be correlated with changes in systemic inflammation in WTC workers with asthma, and whether this is a mechanism underlying worse asthma control.

This study has several strengths including the enrollment of a racially and ethnically diverse cohort of WTC workers with asthma from two large clinical programs. Additionally, this study used the SCID-5, the gold standard for psychiatric interviews in research, alongside the validated PCL-5 tool, to assess PTSD and PTSD symptom severity, respectively. We also used validated methods to measure a comprehensive panel of cytokines in both blood and sputum.

There were, however, some limitations that deserve mention. This study coincided with the COVID-19 pandemic and thus we were limited in our ability to collect sputum samples on all participants due to social restrictions. Moreover, COVID-19 infection can influence cytokine levels and we did not collect information on whether participants had a recent infection. While our study did adjust for participants level of exposure based on exposure to dust cloud, total time spent working at the WTC site and work on the debris pile, we did not assess for other factors that influence extent of exposure such as use of respiratory protection or personal hygiene measures which may have further influenced associations. The lower number of sputum samples limited our study's statistical power and, thus, we cannot exclude that PTSD may have associations with airway levels of specific cytokines. Additionally, our study did not include other WTC exposed individuals beyond WTC workers, thus limiting the generalizability of our results.

In summary, similar to prior studies, we found that WTC workers with PTSD had increased asthma associated morbidity. In this study, we did not find an association between PTSD diagnosis and blood and sputum inflammatory markers among WTC workers with asthma. These findings suggest that other mechanisms, such as perception of PTSD symptoms or other yet to be determined factors, may explain the relationship between PTSD and increased asthma morbidity among WTC workers.

## Supporting information

**S1 Appendix.**
(DOCX)

## Author Contributions

**Conceptualization:** Juan P. Wisnivesky, Adam Gonzalez, Alex Federman, Steven B. Markowitz, Janette M. Birmingham, Paula J. Busse.

**Data curation:** Jyoti Ankam.

**Formal analysis:** Jyoti Ankam, Janette M. Birmingham.

**Funding acquisition:** Juan P. Wisnivesky, Paula J. Busse.

**Investigation:** Juan P. Wisnivesky, Paula J. Busse.

**Methodology:** Juan P. Wisnivesky, Adam Gonzalez, Alex Federman, Steven B. Markowitz, Janette M. Birmingham, Paula J. Busse.

**Resources:** Juan P. Wisnivesky, Janette M. Birmingham, Paula J. Busse.

**Supervision:** Juan P. Wisnivesky, Alex Federman.

**Writing – original draft:** Juan P. Wisnivesky, Nikita Agrawal, Paula J. Busse.

**Writing – review & editing:** Juan P. Wisnivesky, Nikita Agrawal, Jyoti Ankam, Adam Gonzalez, Alex Federman, Steven B. Markowitz, Janette M. Birmingham, Paula J. Busse.

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
