## [Decision Letter · Decision Letter 0]

20 Sep 2023

PONE-D-23-18279Inflammatory Markers in World Trade Center Workers with Asthma: Associations with Post Traumatic Stress DisorderPLOS ONE

Dear Dr. Agrawal,

Thank you for submitting your manuscript to PLOS ONE. After careful consideration, we feel that it has merit but does not fully meet PLOS ONE’s publication criteria as it currently stands. Therefore, we invite you to submit a revised version of the manuscript that addresses the points raised during the review process. Reviewers raised many concerns related to study rationale, design lack of an appropriate control group and issues with statistical analysis.  Please submit your revised manuscript by Nov 04 2023 11:59PM. If you will need more time than this to complete your revisions, please reply to this message or contact the journal office at plosone@plos.org. Please include the following items when submitting your revised manuscript:A rebuttal letter that responds to each point raised by the academic editor and reviewer(s). You should upload this letter as a separate file labeled 'Response to Reviewers'.A marked-up copy of your manuscript that highlights changes made to the original version. You should upload this as a separate file labeled 'Revised Manuscript with Track Changes'.An unmarked version of your revised paper without tracked changes. You should upload this as a separate file labeled 'Manuscript'.

We look forward to receiving your revised manuscript.

Kind regards,

Suresh Pallikkuth

Academic Editor

PLOS ONE

Journal Requirements:

"JPW received honorarium from Banook, PPD, Atea Pharmaceutical, and Prospero and research grants from Regeneron, Sanofi, Axella, and Arnold Consulting. 

PB has received consulting fees from CSL Behring, Takeda, Kalvista, BioCryst, CVS Specialty, Regeneron and research funding from CSL Behring, Takeda, and Kalvista. 

Other authors declare no conflict of interest.

All other authors have no competing interests to declare."

Reviewers' comments:

Reviewer's Responses to Questions

**Comments to the Author**

1. Is the manuscript technically sound, and do the data support the conclusions?

Reviewer #1: Yes

Reviewer #2: Partly

2. Has the statistical analysis been performed appropriately and rigorously? 

Reviewer #1: Yes

Reviewer #2: No

3. Have the authors made all data underlying the findings in their manuscript fully available?

Reviewer #1: Yes

Reviewer #2: Yes

4. Is the manuscript presented in an intelligible fashion and written in standard English?

Reviewer #1: Yes

Reviewer #2: Yes

5. Review Comments to the Author

Reviewer #1: Comments to the Author

Comment: The full form of DSM-5 is not mentioned anywhere in the manuscript

Comment: Line 155: Correct to plasma cytokine levels.

Comment: Line 165: Put space after (J.B.)

Comment: Line 171: Spelling needs to be corrected for Streptavidin-Phycoerythrin.

Comment: As the last part of this study overlaps with the COVID pandemic, did this study consider whether individuals participating in the study was exposed to COVID-19 and whether that would influence the airways and cytokines present?

Comment: Line 201: The P value mentioned in the text and Table 1 are not the same. Which one is correct?

Comment: Line 391, 397, 405 and 411: Correct VEFG to VEGF

Comment: Were any of these samples diluted before testing? RANTES is a marker which needs to be tested after diluting serum/plasma samples using Luminex multiplex assays and usually cannot be tested along with other markers in serum multiplex assays. Were any issues noted with RANTES detection during this study?

Reviewer #2: Overall, this is an interesting work looking to explore whether WTC workers with diagnoses of PTSD and asthma have a specific underlying inflammatory marker profile to explain increased asthma morbidity. Parts of the manuscript, particularly the Introduction and Discussion, do not follow a logical flow to orient the reader to the clinical investigation that was performed. This Reviewer can intuit the line of reasoning in performing some of the secondary analyses, but a more thorough investigation of these factors (particularly the degree of WTC exposure and the outcomes of interest) would strengthen the work substantially in view of the negative findings reported.

Major comments:

1. Please provide rationale for utilizing both blood and sputum samples in the Methods section.

2. Though this reviewer can appreciate why a more thorough statistical analysis was not performed, it seems obvious to investigate some of the other factors collected (such as degree of WTC Exposure) in relation to blood & sputum cytokine levels regardless of PTSD status given that there was no significance in those markers between groups. This would strengthen the paper considerably.

3. Further, this reviewer suggests that if at all feasible, the Authors should include a small control group to understand normative values of this assay for blood & sputum cytokine levels. Please report graphically as a reference in all Tables.

4. It seems theoretically possible that individuals with asthma have upregulation of these values at baseline given that it is a chronic disease. If that is not the case, then justification should be provided in the text of the manuscript in the Discussion.

5. Please further clarify the significance of elevated neutrophils in the sputum cell differential in the Discussion. It is not obvious why this finding is clinically meaningful.

Minor comments:

Line 73: Clarify what is meant by “uncontrolled asthma symptoms” in terms of clinical presentation.

Line 82, Line 87: The role of the cytokines noted in the Introduction are not well-described. Please clarify their roles in relation to PTSD and asthma, respectively.

Line 114: Do the authors mean “opt-in”?

Line 187: Correct to “markers”.

Line 202: Please explain why 29% of individuals were excluded due to lack of blood cytokine measures.

Lines 247-249: Suggest a more thorough review of the literature to include how social and behavioral determinants drive inflammation in chronic conditions. The Authors have not provided a reference for this statement.

Table 2: Please provide reference ranges.

6. PLOS authors have the option to publish the peer review history of their article (what does this mean?). If published, this will include your full peer review and any attached files.

Reviewer #1: No

Reviewer #2: No

---

## [Author Response · Author response to Decision Letter 0]

4 Dec 2023

Dear Editors,

Thank you for the opportunity to resubmit our manuscript “Inflammatory Markers in World Trade Center Workers with Asthma: Associations with Post Traumatic Stress Disorder.” Your feedback was thoughtful and we believe has helped to strengthen our manuscript. Below are our specific answers to your questions as well. 

Thank you

Comment 1. Please ensure that your manuscript meets PLOS ONE's style requirements, including those for file naming. The PLOS ONE style templates can be found at 

Response: Our manuscript now meets PLOS ONE’s style requirements. 

Comment 2. Thank you for stating the following in the Competing Interests section: 

"JPW received honorarium from Banook, PPD, Atea Pharmaceutical, and Prospero and research grants from Regeneron, Sanofi, Axella, and Arnold Consulting. 

PB has received consulting fees from CSL Behring, Takeda, Kalvista, BioCryst, CVS Specialty, Regeneron and research funding from CSL Behring, Takeda, and Kalvista. 

Other authors declare no conflict of interest.

All other authors have no competing interests to declare.”

Response: Although none of the authors’ competing interests alter our adherence to PLOS ONE policies on sharing data and materials, the data used for this study was obtained from a parent study from the WTC Health Program General Responder Data Center at Icahn School of Medicine at Mount Sinai. Because of regulations from the parent study we are not able to share data, even if de-identified. Rather, access to data would have to go through established DCC mechanisms. We have added this to our disclosure statement (lines 35-38). 

Comment 3. In your Data Availability statement, you have not specified where the minimal data set underlying the results described in your manuscript can be found. PLOS defines a study's minimal data set as the underlying data used to reach the conclusions drawn in the manuscript and any additional data required to replicate the reported study findings in their entirety. All PLOS journals require that the minimal data set be made fully available. For more information about our data policy, please see http://journals.plos.org/plosone/s/data-availability.

• What data is uploaded : 

o The data behind the means, SD and other measures reported 

o The values used to build graphs 

o The points extracted from images for analysis 

• How does the process work?

• How is the data secured by journal?

• Who will have access to it in the future?

Response: This project used data from the WTC Health Program General Responder Data Center at the Icahn School of Medicine at Mount Sinai. Because of regulations from the parent study we are not able to share data, even if de-identified. Rather, access to data would have to go through established DCC mechanisms. 

Comment 4. We note that you have stated that you will provide repository information for your data at acceptance. Should your manuscript be accepted for publication, we will hold it until you provide the relevant accession numbers or DOIs necessary to access your data. If you wish to make changes to your Data Availability statement, please describe these changes in your cover letter and we will update your Data Availability statement to reflect the information you provide.

Response: As explained above, due to regulations from the parent study we are not able to share data. We have added a line to explain this in our disclosure statement of the cover letter (lines 35-38).

Response: The corresponding author has included her ORCID iD. 

Response: Please see lines 126-129 where we have clarified that consent was written and added details on the approving IRB committees for this study. 

 

Comments to the Author

1. Is the manuscript technically sound, and do the data support the conclusions?

Reviewer #1: Yes

Reviewer #2: Partly

2. Has the statistical analysis been performed appropriately and rigorously? 

Reviewer #1: Yes

Reviewer #2: No

3. Have the authors made all data underlying the findings in their manuscript fully available?

Reviewer #1: Yes

Reviewer #2: Yes

4. Is the manuscript presented in an intelligible fashion and written in standard English?

Reviewer #1: Yes

Reviewer #2: Yes

 

5. Review Comments to the Author

Reviewer 1 

Comment 1: The full form of DSM-5 is not mentioned anywhere in the manuscript

Response: We have added the full form of DSM-5 to the manuscript. Please see line 159 in the manuscript. 

Comment 2: Line 155: Correct to plasma cytokine levels.

Response: We have corrected this error, please see line 165. 

Comment 3: Line 165: Put space after (J.B.)

Response: We have fixed the spacing issue, please see line 175. 

Comment 4: Line 171: Spelling needs to be corrected for Streptavidin-Phycoerythrin.

Response: We have corrected the spelling error. Please see line 182.

Comment 5: As the last part of this study overlaps with the COVID pandemic, did this study consider whether individuals participating in the study was exposed to COVID-19 and whether that would influence the airways and cytokines present?

Response: We agree that our study did overlap with the COVID-19 pandemic, which is known to influence cytokine release in effected patients. In this study, we did not assess whether patients were actively or recently infected with COVID-19 and the influence this has on sputum and serum cytokines but we look forward to looking at the data. We have added a limitation of COVID-19 on data to our discussion as well (lines 321-322)

Comment 6: Line 201: The P value mentioned in the text and Table 1 are not the same. Which one is correct?

Response: We have adjusted the manuscript to ensure the text and tables have consistent p values. Please see line 227. 

Comment 7: Line 391, 397, 405 and 411: Correct VEFG to VEGF

Response: All spelling errors have been fixed. Please see lines 249, 266, 438, 445. 

Comment 8: Were any of these samples diluted before testing? RANTES is a marker which needs to be tested after diluting serum/plasma samples using Luminex multiplex assays and usually cannot be tested along with other markers in serum multiplex assays. Were any issues noted with RANTES detection during this study?

Response: We agree that RANTES may need to be diluted prior to its measurement in multiplex assays. However, there are several other cytokines which have very low levels of detection and if diluted, may not yield accurate results. We therefore had contacted the manufacture Millipore Sigma prior to running our samples to address this question. The scientific representative from Millipore stated that samples did not need to be diluted for the above reasons and recommended that other investigators have run undiluted samples when measuring RNATES. Additionally, we had other methods in order to assure that the level of RANTES was accurate. These included the standard curve, in which none of the samples run were outside of the curve and that none of the RANTES samples were above the upper limit of detection. Please see lines 178 and 186-187 of the revised manuscript. 

Reviewer 2

Comment 1: Overall, this is an interesting work looking to explore whether WTC workers with diagnoses of PTSD and asthma have a specific underlying inflammatory marker profile to explain increased asthma morbidity. Parts of the manuscript, particularly the Introduction and Discussion, do not follow a logical flow to orient the reader to the clinical investigation that was performed. This Reviewer can intuit the line of reasoning in performing some of the secondary analyses, but a more thorough investigation of these factors (particularly the degree of WTC exposure and the outcomes of interest) would strengthen the work substantially in view of the negative findings reported.

Response: For this study, we categorized participants’ level of exposure as low, intermediate, high or very high based on exposure to the dust cloud, total time spent working at the WTC site and work on the debris piles using a previously published classification. We have included information on level of exposure in Table 1 and adjusted for level of exposure in our analyses as well. We added information on this in our discussion (lines 322-325) 

Comment 2: Please provide rationale for utilizing both blood and sputum samples in the Methods section.

Response: For this study, we examined both blood and sputum samples as prior research has demonstrated both samples provide information on systemic inflammation in asthma [10]. While evaluation of sputum markers are considered an optimal measure of local inflammation, it is often more difficulty to obtain appropriate samples, particularly in the context of the COVID-19 pandemic. Thus, serum samples were additionally obtained (lines 189-193). 

Comment 3: Though this reviewer can appreciate why a more thorough statistical analysis was not performed, it seems obvious to investigate some of the other factors collected (such as degree of WTC Exposure) in relation to blood & sputum cytokine levels regardless of PTSD status given that there was no significance in those markers between groups. This would strengthen the paper considerably.

Response: Our analysis did adjust for WTC exposure and we have clarified this in the manuscript including in our discussion (lines 322-325). 

Comment 4: Further, this reviewer suggests that if at all feasible, the Authors should include a small control group to understand normative values of this assay for blood & sputum cytokine levels. Please report graphically as a reference in all Tables.

Response: This study unfortunately did not include a control group. We did not include a control group as the study was designed to compare the impact of PTSD on asthma. The Multiplex assay has a standard curve which allows validation of cytokine assay levels. 

Comment 5: It seems theoretically possible that individuals with asthma have upregulation of these values at baseline given that it is a chronic disease. If that is not the case, then justification should be provided in the text of the manuscript in the Discussion.

Response: We agree that asthma has both an important local (i.e., airway) and systemic inflammatory components. The study was designed to examine the impact of PTSD (which has a pro-inflammatory component as well) on asthma. 

Comment 6: Please further clarify the significance of elevated neutrophils in the sputum cell differential in the Discussion. It is not obvious why this finding is clinically meaningful.

Response: We have added additional information regarding the importance of elevated neutrophils in asthma as this helps phenotype patients as eosinophilic vs. non eosinophilic and thus has treatment implications. Please see lines 277-279. 

Comment 7: Line 73: Clarify what is meant by “uncontrolled asthma symptoms” in terms of clinical presentation.

Response: We have added additional information on what is defined as uncontrolled asthma symptoms. Specifically, we defined uncontrolled asthma as increased daytime and nighttime respiratory symptoms and frequent rescue medication use (lines 79-80).

Comment 8: Line 82, Line 87: The role of the cytokines noted in the Introduction are not well-described. Please clarify their roles in relation to PTSD and asthma, respectively.

Response: We have clarified the role of cytokines may enhance morbidity in individuals with both PTSD and asthma who are WTC exposed. Please see lines 88-91.

Comment 9 Line 114: Do the authors mean “opt-in”?

Response: We have adjusted this sentence to reflect our recruitment included an initial opt in letter (line 123).

Comment 10: Line 187: Correct to “markers”.

Response: Correction made, please see line 205.

Comment 11: Line 202: Please explain why 29% of individuals were excluded due to lack of blood cytokine measures.

Response: A lack of blood cytokine measures in 29% of individuals was due to various reasons including participant refusal, disruptions to in person study practices by COVID-19 and lack of time during the study visit. 

Comment 12: Lines 247-249: Suggest a more thorough review of the literature to include how social and behavioral determinants drive inflammation in chronic conditions. The Authors have not provided a reference for this statement.

Response: We have included a reference to our recent publication examining the role of symptom perception in WTC workers with asthma and PTSD alongside literature emphasizing social determinants of health in asthma (lines 422-426).

Comment 13: Table 2: Please provide reference ranges.

Response: We have included a statement to figure 2 noting the range of detection for all cytokine assays. Please see line 245.

6. PLOS authors have the option to publish the peer review history of their article (what does this mean?). If published, this will include your full peer review and any attached files.

Do you want your identity to be public for this peer review? For information about this choice, including consent withdrawal, please see our Privacy Policy.

Reviewer #1: No

Reviewer #2: No

---

## [Decision Letter · Decision Letter 1]

10 Jan 2024

Inflammatory Markers in World Trade Center Workers with Asthma: Associations with Post Traumatic Stress Disorder

PONE-D-23-18279R1

Dear Dr. Agrawal,

We’re pleased to inform you that your manuscript has been judged scientifically suitable for publication and will be formally accepted for publication once it meets all outstanding technical requirements.

Kind regards,

Suresh Pallikkuth

Academic Editor

PLOS ONE

Additional Editor Comments (optional):

Reviewers' comments:

Reviewer's Responses to Questions

**Comments to the Author**

1. If the authors have adequately addressed your comments raised in a previous round of review and you feel that this manuscript is now acceptable for publication, you may indicate that here to bypass the “Comments to the Author” section, enter your conflict of interest statement in the “Confidential to Editor” section, and submit your "Accept" recommendation.

Reviewer #1: All comments have been addressed

Reviewer #2: All comments have been addressed

2. Is the manuscript technically sound, and do the data support the conclusions?

Reviewer #1: Yes

Reviewer #2: Yes

3. Has the statistical analysis been performed appropriately and rigorously? 

Reviewer #1: Yes

Reviewer #2: Yes

4. Have the authors made all data underlying the findings in their manuscript fully available?

Reviewer #1: Yes

Reviewer #2: Yes

5. Is the manuscript presented in an intelligible fashion and written in standard English?

Reviewer #1: Yes

Reviewer #2: Yes

6. Review Comments to the Author

Reviewer #1: Authors have adequately addressed my comments and I feel that this manuscript is now acceptable for publication.

Reviewer #2: (No Response)

7. PLOS authors have the option to publish the peer review history of their article (what does this mean?). If published, this will include your full peer review and any attached files.

Reviewer #1: No

Reviewer #2: No

---

## [Editor Report · Acceptance letter]

5 Feb 2024

PONE-D-23-18279R1 

PLOS ONE

Dear Dr. Agrawal, 

I'm pleased to inform you that your manuscript has been deemed suitable for publication in PLOS ONE. Congratulations! Your manuscript is now being handed over to our production team.

Kind regards, 

on behalf of

Dr. Suresh Pallikkuth 

Academic Editor

PLOS ONE